# COVID the Catalyst for Evolving Professional Role Identity? A Scoping Review of Global Pharmacists’ Roles and Services as a Response to the COVID-19 Pandemic

**DOI:** 10.3390/pharmacy9020099

**Published:** 2021-05-04

**Authors:** Kaitlyn E. Watson, Theresa J. Schindel, Marina E. Barsoum, Janice Y. Kung

**Affiliations:** 1EPICORE Centre, Department of Medicine, University of Alberta, Edmonton, AB T6G 2V2, Canada; 2Faculty of Pharmacy and Pharmaceutical Sciences, University of Alberta, Edmonton, AB T6G 2H7, Canada; terri.schindel@ualberta.ca (T.J.S.); mankaruo@ualberta.ca (M.E.B.); 3Oakville Trafalgar Memorial Hospital, Oakville, ON L6M 0L8, Canada; 4John W. Scott Health Sciences Library, University of Alberta, Edmonton, AB T6G 2R7, Canada; janice.kung@ualberta.ca

**Keywords:** COVID-19 pandemic, disaster, pharmacists, pharmacy practice, roles, services, public health, information, medication management, professional role identity

## Abstract

The COVID-19 pandemic requires a range of healthcare services to meet the needs of society. The objective was to explore what is known about the roles and services performed by frontline pharmacists during the first year of the COVID-19 pandemic. A scoping review was conducted of frontline pharmacists’ roles and services during the first year of the COVID-19 pandemic. A medical librarian conducted comprehensive searches in five bibliographic databases—MEDLINE (via Ovid), Embase (Ovid), CINAHL, Scopus, and Web of Science Core Collection for articles published between December 2019 and December 2020. The initial search retrieved 3269 articles. After removing duplicates, 1196 articles titles and abstracts were screened, 281 full texts were reviewed for eligibility, and 63 articles were included. This scoping review presents a conceptual framework model of the different layers made visible by COVID-19 of pharmacist roles in public health, information, and medication management. It is theorized that there is an invisible layer of change representing evolving professional role identity that may influence permanent role change following the pandemic. Thus, the pharmacy profession needs to build upon the lessons and experiences of this global pandemic and not let the momentum of the visible and invisible changes go to waste.

## 1. Introduction

### 1.1. What Does a Pharmacist See Reflected in a Mirror? 

Current literature suggests the pharmacy profession is an extreme case of transformation of professional work [1]. Over the last century, there have been significant evolutionary changes occurring in pharmacy practice in terms of pharmacists’ role definition, pharmacy culture, their perceived place within the healthcare system, and their professional role identity. This is also the case during disasters and emergencies, as it is not widely known what pharmacists’ roles and responsibilities are or should be [2,3]. Although this presents pharmacists with challenges to navigate during crises, it also provides a unique opportunity for pharmacists to create and adapt their roles to meet the needs of society. 

Watson and Colleagues illustrated the evolution of change for pharmacists specific to disasters and emergencies (Figure 1) [4]. Specific disaster events that have occurred in different regions of the world have led to advances in pharmacists’ roles during emergencies. With each passing event, there is greater recognition of pharmacists’ unique contributions to disaster health management. Prior to the events of 9/11 in the United States (US), pharmacists’ roles in relation to disasters were strongly linked to their expertise in logistics and getting medicines from A to B [4]. After the events of 9/11 in 2001 in the US and the severe acute respiratory syndrome (SARS) epidemic in 2003, pharmacists were acknowledged for their drug expertise and contributions in bioterrorism emergencies and pandemics. In 2005, Hurricane Katrina significantly impacted regions of the US and highlighted the valuable pharmacists’ roles in providing clinical services to disaster-affected communities. In 2016, two specific events in different parts of the world—Canada and Australia—ignited the recognition of pharmacists’ essential role in being a first responder in disasters and emergencies providing clinical pharmacy services (e.g., prescribing, assessing, triaging, etc.) [4]. 

### 1.2. Will the COVID-19 Pandemic Follow This Trend and Be the Catalyst for Further Change? 

Hayden and Parkin suggest that pharmacy practice has had to drastically pivot to meet the needs of the pandemic but that it has also “*focused attention on the case for long-awaited professional role evolution*” [5]. They suggest there has been an enduring collective desire within the pharmacy profession for advancing pharmacists’ professional roles [5]. Perhaps, the COVID-19 pandemic provides the fuel for this long-lasting change. Bragazzi and Colleagues agree with this assertion and believe that this pandemic is the beginning of a “new era” for pharmacy practice [6].

Disasters and emergencies present a unique environment and challenge for the healthcare system. There is the accumulation of disaster-specific health needs, existing and chronic illnesses, temporary suspended health services, and overstretched or reallocated healthcare resources [7]. Yet, disasters and emergencies also present opportunities for multidisciplinary and interprofessional collaborations where the traditional healthcare hierarchy is forgotten and the all hands-on deck mentality kicks in. This permeable environment rightfully allows for the rapid expansion of roles and responsibilities, so all healthcare professionals can go where needed and help the most patients, trusting in the autonomy and scope of practice of each profession. For pharmacists, this is evident with the adaptation of their everyday roles (e.g., medication supply, chronic disease management, patient counselling, etc.), but also in the extension to more autonomous responsibilities. For example, during COVID-19, pharmacists simplified hospital in-patient medication regimens (e.g., prescribing therapeutic substitutions, adjusting dosage forms, syncing dose timing in a ward to reduce nursing exposure when administering medications, etc.) [8,9,10,11,12,13,14].

Pharmacists in China were the first to begin describing the adaptations and extensions to pharmacist roles and services that were required to meet the needs of patients in the early phase of the COVID-19 pandemic. In Wuhan China, at the COVID-19 epicenter, field “cabin” hospitals were rapidly built, and pharmacists were required to set up the pharmacy services [15]. Pharmacists were reassigned from other practice settings to assist in providing care in the field hospitals. Various tasks were performed by the pharmacists, including compiling a formulary, procuring medicines, educating patients and other clinicians on the pandemic, evaluating the available evidence to inform clinical decisions for managing COVID-19 infections, monitoring mental health wellbeing, and in an effort to streamline pharmacy services, they outsourced drug information questions from patients and other clinicians to pharmacists in other practice settings (e.g., universities and pharmacy organizations) [15,16,17].

There have been reviews published focusing on pharmacists’ roles during different stages of the COVID-19 pandemic. Visacri and Colleagues identified 11 studies that were published between December 2019 and May 2020 that described pharmacists’ roles during COVID-19 [13]. Merks and Colleagues conducted a similar review but focused on the legal parameters for pharmacists’ roles during the early phase of COVID-19 [14]. Aburas and Alshammari conducted a review of pharmacists’ roles in COVID-19 and compared these to literature on pharmacists’ roles in previous disasters and included articles published up to July 2020 [18]. These reviews and other published studies have provided rapid access to information on pharmacists’ roles as the pandemic unfolded in the early half of 2020 and laid the foundation for this scoping review. The COVID-19 pandemic is continuing to evolve requiring different pharmacy roles and services to meet the needs of society. This scoping review aimed to provide a detailed overview of the roles and services performed by frontline pharmacists during the first year of the COVID-19 pandemic. It provides the building blocks for a sustainable model to be developed for the post-COVID era of pharmacy practice and theorizes that there is a visible and invisible layer that has been unearthed by the pandemic. This study answers the question: What is known about roles and services provided by frontline pharmacists during the first year of the COVID-19 pandemic?

## 2. Materials and Methods

This scoping review followed the methodology outlined by Arksey and O’Malley [19], and it follows the recommendations of the Preferred Reporting Items for Systematic Reviews and Meta-Analyses Statement for Scoping Reviews (PRISMA-ScR) [20]. The completed PRIMSA-ScR checklist is available in the Appendix A. 

### 2.1. Search Strategy

The medical librarian (JK) conducted comprehensive searches in five bibliographic databases on 14 December 2020. They include MEDLINE (via Ovid), Embase (Ovid), CINAHL, Scopus, and Web of Science Core Collection. In order to capture COVID-19-related literature, the COVID-19 search limit was used in Ovid databases, adapted from Ovid’s COVID-19 Tools & Resources for Clinicians. This search strategy was subsequently adapted to other databases. Searches were also limited to articles published from December 2019. A total of 3069 results were retrieved, and when all duplicates were removed, 1196 unique results remained for title and abstract screening. References were imported into Covidence [21], a web-based tool designed to facilitate the screening process for comprehensive reviews. The research team also reviewed the first 200 results from Google Scholar for inclusion, which has been shown to be a reasonable number of results to screen in evidence reviews due to the strong overlap between Web of Science and Google Scholar [22]. Bibliographies from included studies were also reviewed. The complete search strategy is available in the online Appendix A.

### 2.2. Eligibility Criteria

Articles were included if they were written in the English language and they described roles and services provided by frontline pharmacists during the COVID-19 pandemic. All peer-reviewed publication types were eligible for inclusion including reviews, recommendations, and/or guidelines of the pharmacist’s role during the pandemic. Articles that did not describe frontline pharmacists’ roles and services during the COVID-19 pandemic, were written in a language other than English, or published in a predatory journal were excluded from the study. Commentaries that were not peer-reviewed (e.g., editorials) were excluded from this study.

This scoping review focused on the frontline experiences described by pharmacists working during the COVID-19 pandemic. Although other aspects pertaining to pharmacy practice (e.g., proposed roles and services, pharmacy managerial operations, or articles describing the mental health and emotional toll of pharmacists) during COVID-19) are important factors to study and evaluate, they were outside the scope of this scoping review and were not included. 

### 2.3. Data Collection and Extraction

All titles and abstracts were independently screened by two team members (MB and TS). Full-text articles were obtained and reviewed to determine whether the article met the eligibility criteria. If the full texts of the articles were not available in the databases, they were requested through the library interlibrary loan service. Discrepancies in selection were identified, reviewed by a third team member (KW), and resolved through discussion. The article selection process was performed using Covidence, a web-based application designed to help researchers working on systematic reviews, but it may also be applied to other comprehensive reviews such as scoping reviews [21].

Data extraction was performed using a pre-defined template in Microsoft Excel and identified publication date, article type, practice setting, country, objective, roles and services, and the researcher’s summary of the article. A second team member (TS or KW) reviewed 30% of the identified articles for consistency of data collected.

### 2.4. Data Analysis

A content analysis approach was performed on the extracted roles and services highlighted in the articles. Some of the articles described several roles and services that were being performed by frontline pharmacists and were given multiple role and service codes. An inductive approach was used in assigning codes to the roles and services described. The codes were further analyzed into categories and themes using descriptive coding as outlined by Saldana [23]. Meta-analyses were not performed for this scoping review, as the majority of papers were commentaries and narratives. 

## 3. Results

The initial online database search retrieved 3069 articles, and an additional 200 articles were identified from Google Scholar (Figure 2). After removing duplicate items, 1196 article titles and abstracts were screened, 281 full texts were reviewed for eligibility, and 63 articles were included in this study. The full list of the 63 included articles is available in the online Appendix A. The Google Scholar search did not provide any new articles; the articles were either duplicates of the database searches or did not meet the inclusion criteria during title and abstract screening. Full texts were included in this scoping review if they described the experiences, roles, and/or services of frontline pharmacists. The data tables used in this study are available in the online Appendix A.

### 3.1. Study Charactertistics

Table 1 outlines the characteristics of the articles included in the analysis, 33.3% described the hospital practice setting and the majority (74.8%) were written as commentaries. A large proportion of the articles were from the US (31.5%), followed by China (14.3%).

### 3.2. Overall Roles and Services

The complete roles and services table for all 63 articles is provided in the online Appendix A, including the complete list by country and practice setting profiles. There was a total of 570 references to pharmacists’ roles and services described across the 63 articles included in this scoping review. The top 25 roles and services identified in this scoping review are outlined in Figure 3 with definitions provided in the online Appendix A. The most frequently mentioned pharmacists’ roles were interprofessional collaboration [8,10,11,12,13,15,17,24,25,26,27,28,29,30,31,32,33,34,35,36,37,38,39,40,41,42,43,44,45,46,47,48,49,50,51,52,53,54,55,56,57,58,59,60], which was followed by patient education [6,11,12,13,15,16,17,18,24,25,26,28,29,32,33,36,38,39,40,43,45,47,48,49,51,52,55,58,59,60,61,62,63,64,65,66,67,68,69,70] and provider education [6,8,10,11,12,13,15,17,26,27,30,31,32,34,35,37,40,42,43,44,48,49,51,52,54,55,58,59,60,65,71]. This highlights the important services pharmacists provided as reliable evidence-based information sources surrounding the pandemic, treatments, and prevention strategies. Other commonly discussed roles were pharmacists’ redeployment and backfilling of other services during the pandemic (e.g., reassigned to field hospital, COVID-19 wards, procurement, etc.) [8,11,15,17,26,27,31,34,35,44,48,49,50,52,53,54,55,58,60,66], and rounding on COVID-19 patients [8,10,11,12,13,26,27,31,32,35,37,41,42,48,49,50,54,58]. Rounding on COVID-19 patients describes pharmacists’ role in attending in-person or virtual ward rounds to provide their clinical expertise for COVID-19 positive patients.

Unsurprisingly, public health roles being performed by pharmacists during the COVID-19 pandemic were highly reported—COVID screening, reporting, and/or testing [6,11,14,17,18,32,33,39,44,51,52,57,59,64,66,70]; public health messaging [6,11,13,17,25,28,36,43,44,52,55,59,61,67]; managing personal protective equipment (PPE) supplies [6,11,13,14,26,41,44,46,53,65,67,68,70]; combating misinformation [6,25,33,43,52,67]; emotional and psychological support [13,18,44,51,52,59,65,68]; and intriguingly, reporting domestic violence [6,14]. 

Some of the ‘traditional’ roles of pharmacists were reinvigorated and made visible by the COVID-19 pandemic—managing drug shortages [8,10,11,14,15,31,32,33,34,37,48,51,52,70], home delivery [6,11,14,17,18,32,38,40,45,47,61,64,67], compounding [11,14,31,34,35,39,48,49,52,54,67,72], modifying dispensing processes [6,11,14,25,31,38,47,61,68,69,72,73], and simplifying medication regimens [8,10,11,12,13,25,27,31,42,48,49,51,53].

### 3.3. Practice Setting

One-third of the identified articles (33.3%) described pharmacists’ roles and services during the COVID-19 pandemic within the hospital setting (Figure 4). Collaboration and teamwork [8,10,12,26,27,31,34,35,37,41,42,43,48,50,53,54,56,60] and patient care [8,9,10,12,27,31,34,37,41,42,48,53,54,56,60,63,71] roles were emphasized in the hospital practice setting. In contrast, ambulatory care/outpatient clinics were focused on remote service delivery [24,28,29,57,62,66,73] and community pharmacies were focused on medicine supply [14,18,38,40,45,64,68,70,72] and public health [14,18,46,64,68,70] roles (Figure 4). The field hospital articles focused on medicine supply, collaboration and teamwork, and education [15,30,36,55,58], and across the multiple practice settings, patient care, education, public health, and medicine supply were prominent themes [6,11,13,16,17,25,32,33,39,44,47,49,51,52,59,61,65,67,69]. 

The roles performed by pharmacists in the hospital and community practice settings were appreciably different, with only the role of interprofessional collaboration being listed in both of their respective Top 10 (Figure 5). In contrast, when reviewing the articles that described multiple practice settings, evaluating evidence and public health messaging were featured in the Top 10 as important roles performed by pharmacists across all practice settings (Figure 5). The ambulatory/outpatient practice setting articles focused on remote service delivery roles (e.g., drive-through clinics), and the field hospital practice setting focused on field hospital specific operations and medicine supply roles. These other two practice settings account for the 13 articles not depicted in Figure 5. Nursing support was specific to the hospital practice setting and referred to pharmacists’ roles that made them more visible and accessible to nursing staff to support them in providing care to COVID-19 patients. For example, pharmacists anticipated the need and managed the medicine supply for COVID wards and were available to answer nursing queries where necessary [30,31,35,37,44,48,49,55,58,71].

### 3.4. Conceptual Framework Model

This scoping review suggests that the COVID-19 pandemic has made visible the roles and services performed by frontline pharmacists. Globally, these roles are not new but are extensions or adaptations of pharmacists’ existing roles to meet the needs of society during the COVID-19 crisis and can be grouped into three broad categories (Figure 6). While roles are not new (in a global sense), the work of frontline pharmacists is at the local level (at the community/society or team level). Frontline pharmacists are doing new things in their individual context, and they are extending themselves throughout this pandemic.

The conceptual framework model depicts the different layers of pharmacists’ roles and the level of visibility that these roles are pitched at: patients, teams, and community/society. Previously, the most visible roles of pharmacists were those provided to individual patients (e.g., dispensing medicines). Yet, COVID-19 has highlighted and made visible all three layers of pharmacists’ roles: public health, information, and medication management. For example, obtaining PPE supplies, compounding hand sanitizer, and managing drug shortages were major roles in the early phase of the pandemic. In contrast, at the time of writing this scoping review, the COVID-19 vaccine rollout is occurring, and pharmacists are acknowledged as a significant vaccinator and called to be proactive in managing the misinformation surrounding the vaccines. We hypothesize that COVID-19 has radically changed this visibility with the emphasis on pharmacists’ contributions to public health and safety. The impact COVID-19 has had on frontline pharmacist roles is dependent on their individual scope of practice changes; for some, it may have been a catalyst for the creation of a new role, whereas for others, it could have been role expansion. For example, pharmacist prescribing has been normalized in regions by the pandemic with jurisdictions legalizing pharmacists’ roles in refill authorizations or expanding pharmacist prescribing legislation, whereas in other regions, COVID-19 has acted as a catalyst for change in pharmacists’ public health roles (e.g., managing PPE supplies, point-of-care testing). It has also led to adaptations of pharmacists’ roles to remote service delivery or “telepharmacy”, which is expected to be sustained beyond the pandemic.

#### 3.4.1. Medication Management Sphere

The medication management sphere is visible at the individual patient level, and most people would accept and acknowledge these as pharmacists’ roles. COVID-19 has led to a resurgence of these fundamental pharmacists’ roles with pharmacists adapting to remote service delivery as a means of responding and continuing to serve the community [6,8,11,13,14,15,16,17,18,24,25,26,27,28,29,31,33,36,37,38,40,43,44,47,48,54,56,57,58,59,60,61,62,63,64,66,69,73] and meet their medical needs. Other ‘traditional’ medication management roles (e.g., drug supply, managing shortages, and compounding) have also been highly emphasized in the literature during the COVID-19 pandemic [6,8,10,11,12,13,14,15,17,18,25,27,31,32,33,34,35,37,38,39,40,42,45,47,48,49,51,52,53,54,61,64,67,68,69,70,72,73]. 

#### 3.4.2. Information Sphere 

In the middle sphere, information pertains to pharmacists’ roles that are visible to team members and is central to the perception of pharmacists as information professionals or “drug experts”. Many people accept and acknowledge these roles for pharmacists as they share information in the provision of patient care. Yet this has been expanded, as the pandemic has highlighted pharmacists as essential healthcare providers for a global health crisis and identified them as a trusted information source providing valuable input for guidance, education, and policy [6,8,9,10,11,13,15,16,17,25,27,29,30,31,32,34,35,36,37,38,39,41,42,43,44,46,48,49,54,55,56,60,68,74].

#### 3.4.3. Public Health Sphere 

The last sphere shows the emerging pharmacists’ roles in public health and is concerned with protecting and promoting safety. This sphere should be the most visible and recognizable of what roles pharmacists perform, as they are pitched at the community or society level and thus should be visible to all. Previously, this has not been the case, with only a few accepting or acknowledging pharmacists’ public health roles and services. However, the COVID-19 pandemic has shifted the perception of pharmacy practice and accelerated the visibility of the essential pharmacists’ roles performed in a public health crisis [6,11,13,14,17,18,25,26,28,32,33,36,39,41,43,44,46,51,52,53,55,57,59,61,64,65,66,67,68,70]. This is evident in pharmacists being asked to provide COVID-19 screening and testing services, pharmacists broadcasting public health messages, pharmacists providing sanctuary and reporting domestic violence, and pharmacists being required to practice with greater autonomy in the absence of other accessible personnel [11,14,15,27,66]. 

#### 3.4.4. COVID-19 Foundational Context 

Underpinning these pharmacists’ roles is the foundational context, which is essential to understanding the changes that have occurred because of the COVID-19 pandemic but are not discreet to any specific role or service (Figure 6). This understanding emphasizes that although many pharmacists were afforded greater levels of autonomy and responsibilities in the COVID-19 response, this was not done in isolation from other professions but in tandem to meet the health needs of society facing a global crisis. The foundational context was apparent across all countries and practice settings but was visible in different ways. In some countries or practice settings, this was evident in pharmacists prescribing roles, and in others, this was seen in new authority and laws that allowed pharmacists to provide refill authorization or emergency supply services. There was also the acceptance of practicing in discomfort or working in the “gray”. This refers to the nature of disasters and emergencies such as COVID-19, where information is scarce or not yet developed. Pharmacists step up to continue to meet and honor the needs of their patients but without the usual supports of other healthcare providers or a wealth of information at their fingertips to guide their clinical and professional decision making.

## 4. Discussion

### 4.1. What Are Pharmacists’ Roles and Services during COVID-19?

This scoping review looked at the global contributions of frontline pharmacists during the first year of the COVID-19 pandemic. The purpose of this study was to showcase the roles performed by frontline pharmacists and highlight how they responded in different ways to meet the needs of society. Interprofessional collaboration was the role described most often in this study (69.8%, 44/63), and it was the only role emphasized across all five practice settings (Figure 4 and Figure 5) [8,10,11,12,13,15,17,24,25,26,27,28,29,30,31,32,33,34,35,36,37,38,39,40,41,42,43,44,45,46,47,48,49,50,51,52,53,54,55,56,57,58,59,60]. This role highlights the importance of multidisciplinary teamwork during disasters and emergencies such as the COVID-19 pandemic. It also emphasizes that although pharmacists were afforded greater levels of autonomy, this was not done in isolation from other professions but in tandem to meet the health needs of society. The study findings are comparable to the reviews that were conducted about pharmacist roles during the first half of 2020. Visacri and Colleagues found an overabundance of articles from the US and China, which was echoed in this scoping review [13]. The commentary published by the International Pharmaceutical Federation [75] identified a similar trend to this scoping review with prominent themes of pharmacists’ roles in response to COVID-19 being a shift to remote service delivery and pharmacists’ involvement in COVID-19 testing. 

Nadeem and colleagues suggest there has been a paradigm shift of pharmacists’ roles to a focus on patient care, as a trusted information source with greater autonomy because of COVID-19 [76]. This sentiment is echoed by Hayden and colleagues, who believe the pandemic will push the pharmacy profession into a “new era” of pharmacy practice [5]. This scoping review supports these ideas and presents a conceptual framework model of how the roles performed by frontline pharmacists in the context of a pandemic could be the catalyst of change for this “new era” of pharmacy and perhaps lead to a new equilibrium for sustainable pharmacists’ professional role changes. This conceptual framework model and the roles highlighted in this scoping review about the COVID-19 pandemic are reflected in previous studies conducted by Watson and Colleagues about pharmacists’ roles in natural and anthropogenic (i.e., manmade) disasters. These studies presented a similar conceptual framework model of the four practice areas that pharmacists’ roles can be grouped: logistics, patient care, governance, and public health [7,77,78]. This is mirrored in this scoping review with the same practice areas being highlighted using slightly different language: medicine supply, patient care, medication management, policy and governance, and public health.

### 4.2. What Are the Public Implications of These Roles?

The roles described in this scoping review are not new to the pharmacy profession or outside their scope of practice but were made more visible by the pandemic and societal needs. For example, managing drug shortages or compounding medicines are established roles of pharmacists, but the unique challenges of the pandemic saw the resurgence of the importance of these roles, bringing them to the forefront of the public’s attention about what pharmacists do. Similarly, COVID-19 catalyzed the necessity for pharmacists to perform public health roles to increase the available healthcare resources and personnel (e.g., COVID-19 screening, testing, and public health messaging) [6,11,13,14,17,18,25,28,32,33,36,39,43,44,51,52,55,57,59,61,64,66,67,70]. Pharmacists were also integrated in other public health initiatives such as reporting domestic violence as pharmacies were recognized as a community hub and safe haven where victims of domestic violence could use the codeword “Mask 19” with a pharmacist at the counter of a pharmacy to report domestic abuse or violence, even with their abuser present [6,14]. 

Pharmacists’ skills in critical appraisal of evidence were accelerated to meet the overwhelming need to provide accurate and informative guidance to patients, other healthcare providers, and the general public [6,8,10,11,12,13,15,16,17,18,24,25,26,27,28,29,30,31,32,33,34,35,36,37,38,39,40,42,43,44,45,47,48,49,51,52,54,55,58,59,60,61,62,63,64,65,66,67,68,69,70,71]. As noted, while none of these roles were new for pharmacists, the creative methods and public visibility of these roles was altered. Pharmacists were using resourceful methods to deliver public health advice and health information. For example, pharmacists were broadcasting on radio and television to provide COVID-specific public health messaging and evidence-based information on treatments and public health measures [13,28,36,37,42,67]. Another major change to pharmacists’ roles that was illuminated in the public eye was the shift to remote service delivery [24,28,29,57,62,66,73]. Remote patient management and home care became the norm with pharmacists performing counseling or monitoring services via virtual platforms and ‘drive-through’ clinics for patient monitoring. It is anticipated that these “telepharmacy” services will continue to grow in popularity with the global technology shift beyond the pandemic. 

### 4.3. What Do Pharmacists See in the Mirror? 

This scoping review can be used by frontline pharmacists to look at what others are doing. While these roles are not new in a global sense, the work performed by frontline pharmacists is at the local level (community/society or team). Pharmacists are doing new things in their individual context and they are extending themselves throughout this pandemic. Pharmacists’ experiences with the roles and changes in roles while responding to the needs of patients and society during the first year of the COVID-19 pandemic may have implications for their professional role identity. The relationship between professional roles and identity are intimately related. Roles represent a social prescription for behavior, whereas identity is an internal self-understanding of the professional role [79]. Professional role identity represents how professionals see themselves in relation to their professional roles: who they are and how they should act [80,81]. Professional role identity is dynamic. Changes in work role are among the many contributors to changes in professional role identity. Other contributors include actual change in behavior [82], search for meaningful roles [80,83], alignment between work and personal values [81], interactions with other professionals [81,83,84,85], professional organizations [86,87], and past experiences [88]. Role models [83] and colleagues provide important information to support the process of adopting new roles and changing professional role identity [80,85]. The collated global roles highlighted in this scoping review are a mirror in which pharmacists can see their reflection of their various roles and evolving professional role identity. 

COVID-19 presents an opportunity for pharmacists to see themselves through the essential roles that they have performed during the COVID-19 pandemic. Pharmacists have been relied upon for every phase of the pandemic and continue to be called to provide healthcare to society. Through the lens of professional role identity [80,81], perhaps the prolonged nature and demands of COVID-19 has been the catalyst to the new equilibrium of pharmacy practice change and pharmacists’ professional role identity. 

### 4.4. What Does It Mean for the Future of the Pharmacy Profession?

Watson and colleagues identified in previous disaster events that pharmacists have not received fair recognition for their contributions, and once the event has passed, so too does the inclusion of pharmacists in disaster health management [3,4,7]. However, there seems to be a distinct difference with the COVID-19 pandemic. COVID-19 is the first disaster in over a century that has simultaneously impacted the entire globe and infected all four corners of the world. This has led to a universal recognition of pharmacists as essential members of the healthcare workforce, with pharmacies being recognized as one of the main community healthcare services still accessible to patients in the community. COVID-19 has made visible the changes to pharmacists’ roles as illustrated in this scoping review. However, the invisible changes of how COVID-19 has impacted pharmacists’ professional role identity are unknown. 

The pharmacists’ roles identified in this study are a model for reimagining pharmacists’ roles and pharmacy practice. Changes in the work of pharmacists associated with the COVID-19 pandemic set the stage for monitoring long-lasting professional role changes and understanding evolving professional identities. This poses the question: What will be the “new era” for pharmacy? We hypothesize that COVID-19 is the catalyst for sustainable change and the next evolutionary step for better integration of pharmacists into disaster health management [4]. As Paudyal and Colleagues explained, COVID-19 has removed the healthcare pecking order, “*…the same PPE and color of uniforms being worn during the pandemic leveled the hierarchy and therefore everyone was ‘seen as equal’*” [52]. This leveling of the healthcare playing field opens the opportunity for pharmacists to re-imagine their roles and professional role identity. 

This study is but a glimpse into the work of frontline pharmacists to ensure people get their medicines, have supplies to prevent the spread of the virus, have reliable and evidence-based information, and receive ongoing care and support. Further research is required to understand the next chapter in the COVID-19 pandemic saga (e.g., vaccinations, recovery, preparation for future disasters and emergencies). Further research is needed to explore pharmacists’ experiences of COVID-19 and thus fully understand and address pharmacists’ evolving professional role identity. 

### 4.5. Limitations

This scoping review included articles that were published in 2020; there may have been studies that were conducted during or were about the first year of the pandemic but were not published online before 14 December 2020 and thus were not included in the analysis. This scoping review was limited to peer-reviewed literature and articles published in English. We believe this scoping review is generalizable, as it spans 18 countries, five pharmacy practice settings, and due to the necessity of rapidly available published literature during the early phase of the pandemic, many articles were written in English to reach a wider audience. Due to the wealth of articles published on pharmacists and COVID-19, we narrowed our focus to specifically capture the roles being performed by frontline pharmacists. We made the decision to exclude proposed roles during the data extraction process. This decision was made as there were articles that hypothesized about what roles pharmacists could do in response to COVID-19, but to answer our research question, we wanted to specifically know about the actual experiences and what roles were being actioned by pharmacists. We also excluded articles about operations/managerial roles and the mental/emotional toll COVID-19 has had on pharmacists. While these aspects of pharmacy practice are important, they were beyond the scope of this study. We also excluded gray literature; while we acknowledge there is likely more information being discussed regarding pharmacists’ roles not captured in peer-reviewed published articles, the decision was made to focus the papers on peer-reviewed information. 

## 5. Conclusions

### Is the COVID-19 Pandemic a Catalyst for Change in Pharmacy Practice? 

This scoping review presents pharmacists’ roles being performed on the frontline during the first year of the COVID-19 pandemic and highlighted the different layers made visible by COVID-19 of pharmacist roles in public health, information, and medication management. It is theorized that there is an invisible layer of evolving pharmacist professional role identity being exposed by the COVID-19 pandemic. Thus, the pharmacy profession needs to build upon the lessons and experiences of this global pandemic and not let the momentum of the visible and invisible changes to pharmacists’ roles and identity go to waste. 

## Figures and Tables

**Figure 1 pharmacy-09-00099-f001:**
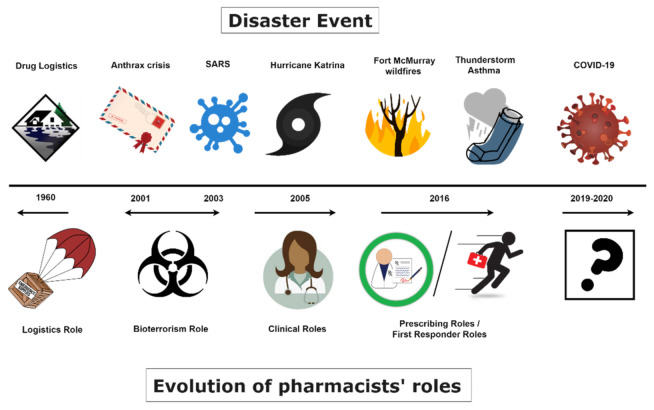
Illustration of the evolution of pharmacists’ roles in disasters since 1960, published in May 2020 in the Canadian Pharmacists Journal and reproduced with permission [4].

**Figure 2 pharmacy-09-00099-f002:**
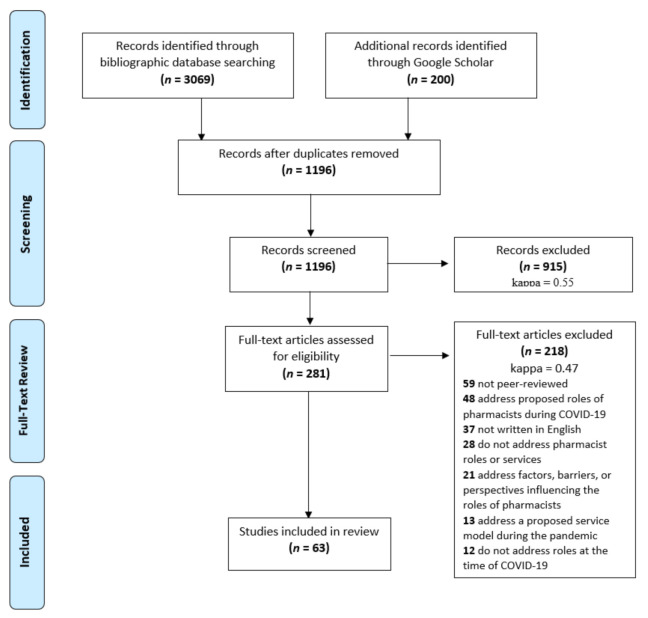
Prisma Flow Diagram.

**Figure 3 pharmacy-09-00099-f003:**
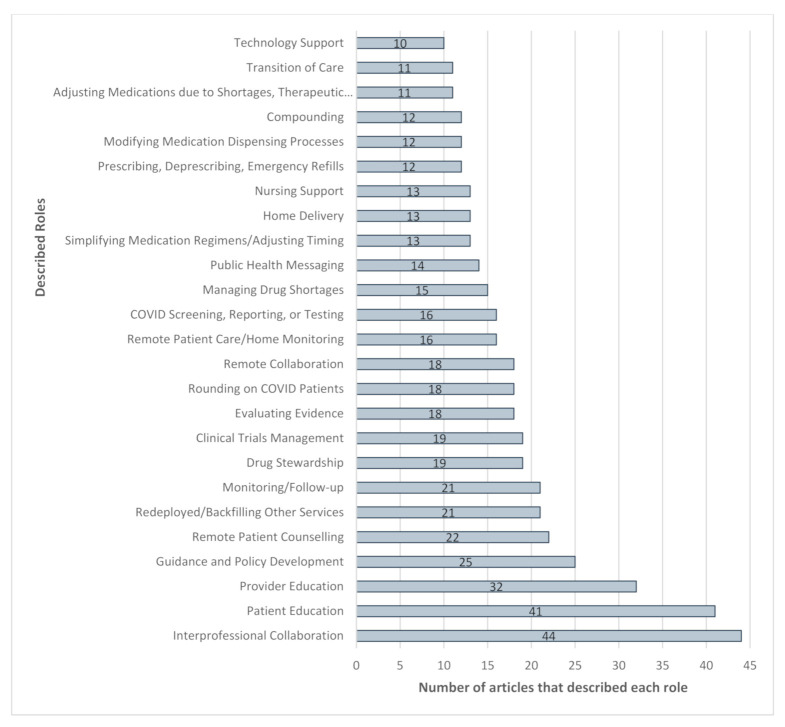
Top 25 ranked roles and services performed by pharmacists during COVID-19. *n* = number of articles that described each role (total = 63 articles).

**Figure 4 pharmacy-09-00099-f004:**
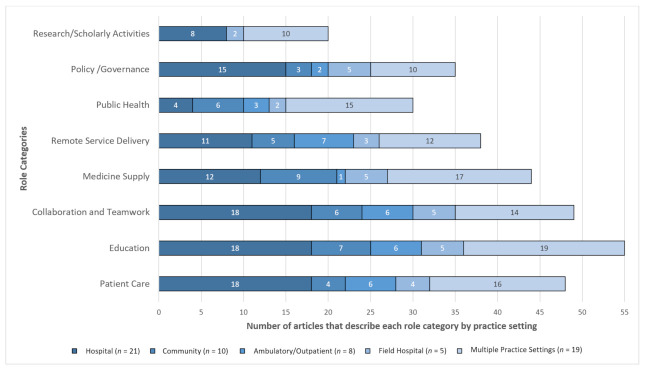
Role categories by practice setting. Each role category consists of several roles (e.g., medicine supply included nine roles, public health included six roles). The full list of articles and roles for each practice setting is available in the online Appendix A.

**Figure 5 pharmacy-09-00099-f005:**
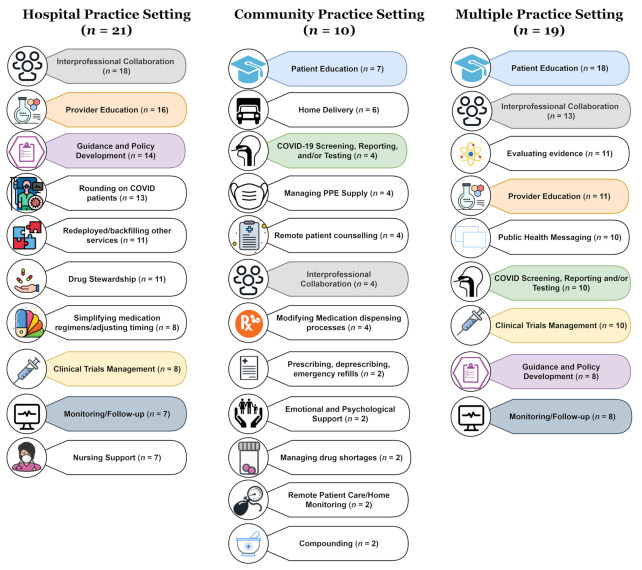
Comparison of roles performed by frontline pharmacists during COVID-19 in the hospital, community, and across multiple practice settings (*n* = number of articles).

**Figure 6 pharmacy-09-00099-f006:**
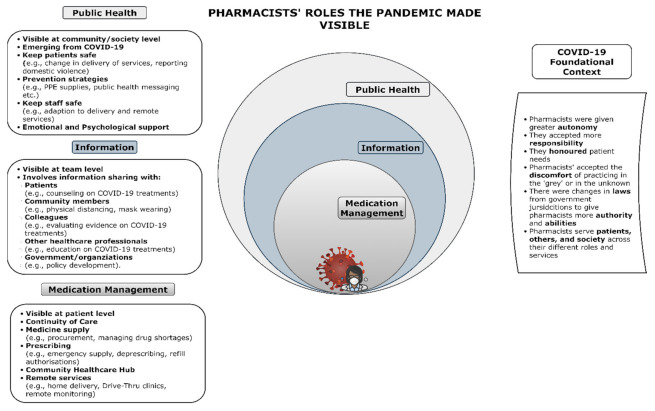
Conceptual framework model of pharmacists’ role change during COVID-19 pandemic.

**Table 1 pharmacy-09-00099-t001:** Article characteristics.

Practice Setting	% (*n* = 63)	Country	% (*n* = 63)
Ambulatory Care and Outpatient Clinics	12.7% (8)	Australia	1.6% (1)
Community	15.9% (10)	Brazil	1.6% (1)
Field Hospital	7.9% (5)	Canada	3.2% (2)
Hospital	33.3% (21)	China	14.3% (9)
Multiple practice settings	30.2% (19)	Ethiopia	1.6% (1)
**Methods Described**	**% (*n* = 63)**	France	1.6% (1)
Commentary	74.6% (47)	Indonesian	1.6% (1)
Literature Review	7.9% (5)	Jordan	1.6% (1)
Qualitative—Interview	6.3% (4)	Malaysia	1.6% (1)
Quantitative—Content Analysis	1.6% (1)	Netherlands	1.6% (1)
Quantitative—Observational Study	4.8% (3)	Pakistan	3.2% (2)
Quantitative—Survey	4.8% (3)	Saudi Arabia	4.8% (3)
**Published Online Month 2020**	**% (*n* = 63)**	Spain	6.3% (4)
February	1.6% (1)	Taiwan	3.2% (2)
March	1.6% (1)	Thailand	1.6% (1)
April	7.9% (5)	United Arab Emirates	3.2% (2)
May	12.7% (8)	United Kingdom	3.2% (2)
June	19% (12)	US	31.5% (20)
July	14.3% (9)	Multiple Europe Countries	1.6% (1)
August	6.3% (4)	Multiple Countries/Worldwide	11.1% (7)
September	3.2% (2)		
October	20.6% (13)		
November	7.9% (5)		
December	4.8% (3)		

## Data Availability

The data presented in this study are available in the Appendix A.

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
