# Peer review of "COVID the Catalyst for Evolving Professional Role Identity? A Scoping Review of Global Pharmacists’ Roles and Services as a Response to the COVID-19 Pandemic"

_pharmacy, 2021, doi:10.3390/pharmacy9020099_

Round 1

Reviewer 1 Report

Executive Summary

The manuscript titled “COVID the Catalyst for Evolving Professional Role Identity? A Scoping Review of Global Pharmacists’ Roles and Services as 3 a Response to the COVID-19 Pandemic” is a systemic review of the current literature regarding pharmacists’ role change in 2020. Overall, the manuscript is well written with robust discussion and strong logic. Authors may perform minor revisions for clarification purposes.

Major Comments

Section 3.4. Continent

Since registered pharmacist has different requirements in different countries, it may be inappropriate to compare them directly without proper discussion. I know that China and the US have huge differences in R.Ph. Authors may delete section 3.4 or write one paragraph in the discussion section to provide more details.

Minor Comments

In Figures 3 to 6, it would be better to have a consistent description of the data. Please see the following:

  • Figure 3: Number of times each role was described
  • Figure 4: Number of Roles described for each Practice Setting
  • Figure 5: n= number of articles
  • Figure 6: Number of Roles described for each Continent

Is this number the number of articles? Or simply how many times the role has been mentioned in one article? It will make a huge difference in numbers if using different standards. Please clarify.

Author Response

Thank you to the reviewer for their insightful comments and perspective. We agree there is difficulty in providing comparisons across countries or continents due to the variation of pharmacists’ scope of practice and legislation. Upon reflection during the peer-review process, we have decided to remove section 3.4 as we don’t want to detract from the overall message of the paper and study, and we agree that continents like the Oceania region is underrepresented for comparisons.  

For the figures, thank you for highlighting the inconsistency. Our apologies for Figure 3, it is the number of articles that mentioned that role and not the number of times the role was mentioned in articles, Total n=63 articles. This has been corrected in the axis label and made clear in the figure caption. Figure 4 has been updated to reflect number of articles for consistency. Figure 6 and section 3.4 has been removed.

Reviewer 2 Report

Thank you for your review. The impact of COVID-19 has been widespread and often unequal across nations. Perhaps those nations not publishing in the article time frame may not have published so frequently before COVID-19 or they may have been significantly differently impacted and been unable to do so. Their voices may however offer insights in time so perhaps looking to 2021-2022 may identify more articles and different roles and services.

Author Response

Thank you to the reviewer for their detailed review and response. We agree this scoping review has ongoing relevance beyond this specific disaster event. We love the onion peeling analogue as this is what we were hoping would be portrayed through this paper. Thank you for the comment about change usually being a comparison to what came before. We made the decision to try and capture what came before in the introduction with the description of the evolution of pharmacist roles in previous disasters. We have expanded this section to provide more context for Figure 1 and outline the disaster events and roles that have evolved the pharmacy identity in terms of disaster health management (lines 43-52).

We agree that the article highlighted by the reviewer “Langran C, Willis S, Hughes L, Mantzourani E, Hall K. Intra and Interprofessional working: how have pharmacists’ working practices changed during the COVID-19 pandemic? International Journal of Pharmacy Practice. 2021; 29(Supplement_1):i35-i6” would be interesting to provide some background and insight into the changes that have occurred during COVID-19, but unfortunately it is an abstract and more information on the study and findings are needed to provide a comprehensive background.

Thank you for the feedback about having more context for an international audience. Upon reflection of our manuscript and the feedback received, we agree that some regions like the Oceania region were underrepresented in their data. One of the authors is Australian and knows that their COVID-19 situation is vastly different with no community spread, compared to other regions like North America, and the Pacific Islands were less impacted during 2020 but are beginning to become overwhelmed with the 2nd/3rd wave that is raging now in 2021. We anticipate that more research will come from these underrepresented regions as the pandemic continues and in the recovery phase. We have made the decision based on the feedback received to remove section 3.4 and the comparison between continents as this is not a major focus of the article and we do not want to detract from the key message about the breadth of pharmacists’ roles and services performed during COVID-19 and how it has impacted the internal and external identity of pharmacists.  

For the results section, we have amended the figures to be consistent, so they all reflect the number of articles that describe the role. We were intentionally broad in the role descriptions used for a specific role to capture the variations across different practice standards. For example, rounding on COVID-19 role code was used whenever a paper spoke about pharmacists’ role in providing their expertise specific to COVID-19 positive patients, this included in-person attendance of ward rounds, video consults, etc.) this has been added to the results section where this role is discussed (line 197-199) and it is also captured in the definitions for these roles provided in the online supplementary material. We have also elaborated on the role description for ‘Nursing support’ in relation to Figure 5 (line 241-246).